# Large-Area Thermal Distribution Sensor Based on Multilayer Graphene Ink

**DOI:** 10.3390/s20185188

**Published:** 2020-09-11

**Authors:** Tomi Koskinen, Taneli Juntunen, Ilkka Tittonen

**Affiliations:** Department of Electronics and Nanoengineering, Aalto University, P.O. Box 13500, FI-00076 Aalto, Finland; taneli.juntunen@iki.fi (T.J.); ilkka.tittonen@aalto.fi (I.T.)

**Keywords:** graphene, thermoelectric, sensor, flexible

## Abstract

Emergent applications in wearable electronics require inexpensive sensors suited to scalable manufacturing. This work demonstrates a large-area thermal sensor based on distributed thermocouple architecture and ink-based multilayer graphene film. The proposed device combines the exceptional mechanical properties of multilayer graphene nanocomposite with the reliability and passive sensing performance enabled by thermoelectrics. The Seebeck coefficient of the spray-deposited films revealed an inverse thickness dependence with the largest value of 44.7 μV K^−1^ at 78 nm, which makes thinner films preferable for sensor applications. Device performance was demonstrated by touch sensing and thermal distribution mapping-based shape detection. Sensor output voltage in the latter application was on the order of 300 μV with a signal-to-noise ratio (SNR) of 35, thus enabling accurate detection of objects of different shapes and sizes. The results imply that films based on multilayer graphene ink are highly suitable to thermoelectric sensing applications, while the ink phase enables facile integration into existing fabrication processes.

## 1. Introduction

Temperature sensing is pivotal in a wide variety of applications, ranging from monitoring environmental conditions [1] to food safety [2] and medical care, where changes in body temperature are used to monitor the state of health [3,4,5]. In addition to gaining information on physiological conditions, body temperature may be utilized to detect touch and gestures by exploiting thermoelectrics [6]. Such an approach provides an alternative to conventional resistive and capacitive sensing technologies [7] for realizing tactile human-machine user interfaces. For successful integration into these applications, the thermoelectric sensors are required to possess a light and flexible form factor, as well as to be environmentally friendly [8,9,10].

Graphene, the two-dimensional allotrope of carbon, has emerged as an active material in a range of sensor architectures due to its remarkable mechanical durability [9,10,11,12] and high sensitivity to various stimuli from the environment [12,13]. Films of graphene can be fabricated in large area either by chemical vapor deposition (CVD) or via liquid-phase processes, thus fulfilling the physical size requirements associated with tactile user interfaces. Recently, solution-processed few-layer graphene has been reported to provide a thermoelectric response with Seebeck coefficient S > 80 μV K^−1^ [14] and power factor PF = S^2^σ > 600 μW m^−1^ K^−2^ [15,16], rendering it intriguing for thermal sensing applications as well.

Despite the favorable characteristics, thermoelectric applications exploiting graphene remain scarcely reported. Previous reports considering the utilization of graphene as an active material for thermoelectric devices include the exploitation of both CVD-grown and solution-processed graphene [16,17,18,19,20] often in combination with either carbon nanotubes (CNTs) or conductive polymers [15,21,22,23,24,25]. Nevertheless, only a few of the proposed approaches possess a large-area architecture suitable for detecting distributions in temperature [19,26].

The possibility to produce graphene from bulk graphite via a range of liquid-phase methods, e.g., ultrasonic-assisted liquid phase exfoliation (UALPE) [27,28], further permits scaling up the fabrication process into mass-production volumes [14]. Moreover, the liquid phase provides additional advantages in terms of device fabrication; functional ink formulations are inherently suitable for printing processes, thus enabling lithography-free selective deposition [29,30]. Furthermore, these inks are also suited for spray-coating, where high throughput can be achieved by exploiting roll-to-roll techniques [31,32].

While different realizations of flexible temperature sensor arrays have been reported previously [33,34,35,36,37,38,39], they are mainly based on thermistors and often either require external power or hinder from complex fabrication processes. Here, we report a large-area thermal distribution sensor utilizing the thermoelectric character of multilayer graphene ink. The sensor design is based on a distributed thermocouple architecture relying on a single thermoelectric thin film acting as a common leg for an array of thermocouples [6].

The ink approach used to fabricate the graphene film provides a novel and accessible way of preparing flexible and mechanically durable active layers for large-area thermoelectric sensors. Compared to previous temperature-sensing approaches exploiting graphene, the proposed sensor is simple and inexpensive to fabricate, while still comprising multiple pixels, a feature that is lacking in many alternative designs utilizing the thermoelectric effect. The device is demonstrated both as a touch sensor and a real-time thermal distribution sensor, making it suitable for a wide range of applications, for example, in wearable and flexible electronics [40,41].

## 2. Materials and Methods

The graphene inks were prepared through a process similar to that reported previously [20]. First, bulk graphite (300 mg, 100 mesh, Sigma-Aldrich, Finland) was mixed with isopropyl alcohol (30 mL) and polyvinylpyrrolidone (PVP, 4.5 mg, m.w. 10,000, Sigma-Aldrich, Finland). The addition of PVP inhibited aggregation of the exfoliated dispersion. The solution was immersed in a temperature-controlled ultrasonic bath for 8 h at a temperature below 30 °C, resulting in the exfoliation of the graphitic species. The graphene suspension was centrifuged at 1050 RCF for 1 h to separate the heavy non-exfoliated graphite particles from the multilayer graphene flakes, after which the supernatant was collected by extracting the top 80% of the centrifuged mixture. The obtained solution was used as is and was found shelf-stable for extended periods of several weeks with no observable deterioration of film properties once deposited.

The thermal distribution sensor was fabricated as follows: The prepared multilayer graphene ink (45 mL) was spray-deposited onto Kapton polyimide substrate over an area of 6 × 6 cm^2^. The film was then thermally annealed at 375 °C for 30 min to remove the weakly conductive PVP stabilizer from the films, resulting in an order of magnitude increase in electrical conductivity. Following the annealing treatment, 250 nm Al_2_O_3_ dielectric and 30/250 nm Ti/Au readout leads were deposited by e-beam evaporation (Edwards E306A) through a shadow mask. External wiring was connected by the use of either silver epoxy (CW2400, CircuitWorks, RS Components Finland) or conductive carbon paste (product no. 16057, PELCO, Ted Pella USA, Redding, CA, USA). The produced thermoelectric signals were analyzed with National Instruments SignalExpress software via the NI cDAQ 9213 data acquisition module.

Structural analysis of the device was carried out using Zeiss Supra 40 scanning electron microscope (SEM), Bruker DekTak XT surface profilometer and WiTec Alpha 300 RA+ Raman spectroscopy equipment. Seebeck coefficients and resistivities of the samples were determined using a Linseis LSR-3 measurement system equipped with a thin film adapter. Five different temperature gradients were used for the measurement and carried out at an average temperature of 34 °C in an inert He atmosphere. The measured temperature difference and voltage values were analyzed using linear regression, and the Seebeck coefficient was obtained from the slope of the fit. Charge carrier mobilities and concentrations of the samples were obtained with Ecopia HMS-5300 Hall measurement system using van der Pauw contact geometry. An infrared camera (Testo 865) was used to acquire reference heat maps for the thermal distribution characterization. The electrical resistance of a sample film was measured under controlled deformation using a Fluke 289 multimeter, as well as under controlled humidity in a closed chamber using a Keithley 2602B source meter.

The device architecture was reproduced in finite-element method (FEM) simulation software (COMSOL Multiphysics). The heat directed onto the device surface was modelled as a boundary heat source with a Gaussian spatial temperature distribution. The temperature rise produced by the boundary heat source was limited to 1.2 K, replicating the finite temperature rise resulting from the touch of a finger. Further details concerning simulation parameters and modelling results are included in Appendix A

## 3. Results

### 3.1. Sensor Architecture and Structural Properties

Figure 1a presents both the exploded-view drawing and top view of the thermal distribution sensor architecture. The sensor operates as follows: The deposited multilayer graphene film connects to electrical ground at the edge. In addition, the graphene film is in contact with the ends of 16 readout leads arranged in a four-by-four matrix. The Al_2_O_3_ dielectric layer separates the remainders of the readout leads from the graphene film. Each readout lead forms an effective thermocouple with the graphene film acting as a common leg. When the contact between a readout lead and the graphene film is heated, a temperature gradient forms between the contact point and the reference ground, resulting in a thermoelectric voltage determined by the effective Seebeck coefficient of the graphene film and the readout lead. As a result, the spatially distributed effective thermocouples act as a sensor array accepting input signals of thermal origin. Since the voltage generation takes place within a single open circuit, the electrical readout wiring of the device is remarkably simplified compared to arrays of individual sensors, requiring only half of the electrical wiring.

Figure 1b,c show photographs of the device before and after the deposition of readout leads. The device was mechanically durable and remained semitransparent, owing to the nanoscale film thickness, as displayed in Figure 1c. Figure 1d presents an SEM micrograph of the film surface morphology, showing stacking of the flakes and the characteristic flake size in the range of hundreds of nanometers. The film was confirmed to have an average thickness of 95 nm as determined from a profilometer measurement on a reference film deposited on glass. Raman spectroscopy was exploited to confirm that the multilayer graphene retained its structural properties [42] and pristine crystalline quality in spite of the annealing treatment performed to remove the dielectric PVP component, as evidenced by the virtually unchanged Raman spectra with I_D_/I_G_ ratio of 0.26 before and after the treatment presented in Figure 1e [20,43]. Moreover, the shape of the 2D peak in the spectra provided a clear signature of the multilayer character of the flakes. This is further supported by the I_2D_/I_G_ ratio of 0.36 acquired from the post-annealing spectrum [44].

### 3.2. Thermoelectric Properties

To investigate the effect of film thickness on the sensor thermoelectric properties, 4, 8, 12 and 16 mL of ink were deposited on 2 × 3 cm^2^ areas. After annealing, the samples were characterized as summarized in Table 1. The film of the 8 mL sample corresponded to the presented sensor device with a thickness of ~100 nm. The films expressed p-type character typical of graphene films annealed in air, which may be attributed to doping by atmospheric oxygen and moisture [45,46]. As the sample thickness increased, hole mobility increased and resistivity decreased, likely due to improved flake interconnectivity in the nanoporous structure stacked via van der Waals forces. Simultaneously, the measured Seebeck coefficients ranged from 37.4 to 44.7 μV K^−1^ with decreasing film thickness, in sound agreement with the peak voltages obtained in touch sensing experiments. The obtained hole concentration values agreed well with previously reported values for few-layer graphene flakes fabricated via a similar process [20]. The decreasing trend as a function of increasing film thickness may be assigned to the lower degree of atmospheric doping deeper in the film. Concurrently, the observed hole mobility was roughly 1.5-fold compared to the previous results, showing a lower resistivity and an improved power factor ranging from 22.1 to 29.3 μW m^−1^ K^−2^ depending on the sample. The results were not largely dependent on film thickness. However, thinner films appeared preferential for sensor application due to both a higher Seebeck coefficient and higher optical transparency, which promotes applications where transparency is important from either a practical or aesthetic standpoint, such as wearable or window applications. Additionally, to observe any possible changes in electrical properties due to deformation of the graphene film, the electrical resistance of a sample film was measured with the film bent to different bending radii, showing only minor changes at the smallest bending radii and thus highlighting the excellent mechanical durability of the films. The relative change in resistance is presented in Appendix A. Furthermore, the effect of humidity on the film resistance is presented in Appendix A.

### 3.3. Transient Response

The transient performance of the device was characterized by carrying out a series of experiments, where the pixels were gently touched by a finger wearing a nitrile glove. Figure 2a,b present the obtained transient touch responses for single touch and consecutive touches, respectively. The transient signals consisted of two distinct time intervals, namely, the linearly rising behavior immediately after the beginning of a touch and the more slowly rising contribution emerging near the peak of the signal. Typical signal rise and decay times resulting from a 2 s touch were approximately 150 ms and 15 s, respectively, as presented in Figure 2a. A FEM simulation describing the device architecture was used to reproduce the touch response. The simulated signal accurately followed the single-touch signal presented in Figure 2a (dashed line). The FEM simulations confirmed that the temperature gradient available in touch applications is usually in the order of 1 K, providing a large-enough thermoelectric signal for practical use.

Figure 2b shows the response under consecutive stimulation. The peaks corresponding to individual touches are easily observed despite the device not having enough time to cool down to its initial state. Moreover, Figure 2c presents the response to brief one-second long touches, produced by a heated rubber tip at six different temperatures ranging from 30 °C to 90 °C, measured on the outer surface of the tip. The inset shows the linear relation between the acquired signal peak voltages and tip temperatures, indicating suitability to applications in a broad range of temperatures. We note that the generated signal was smaller than that suggested by the Seebeck coefficient of the film and the temperature of the rubber tip. This was due to short contact time and high interfacial thermal resistance between the rubber tip and the sensor surface.

### 3.4. Thermal Distribution Mapping

Thermal distribution mapping was demonstrated in two separate experiments. First, the device was attached onto the outer surface of a beaker containing warm water at a temperature of 36 °C. The signal was recorded after 15 s, sufficient for the voltage reading to reach a steady state. Figure 3a presents the sensor attached onto the surface of the 1 L beaker, demonstrating the flexibility of the device. A two-dimensional map exploiting natural neighbor interpolation between the neighboring pixels presents the captured signals. We note that the data represent the raw thermoelectric output of the sensor and no further calibration of the individual pixels was carried out. Thus, the sensor provided excellent uniformity in terms of its output. Analysis of error between different legs has been included in Appendix A. The sensor detected the surface level and the temperature of the water inside the beaker accurately, with the actual water surface level residing between the two topmost pixel rows. Responses to different water levels and temperatures are presented in Appendix A.

Figure 3b presents the second experiment demonstrating shape detection based on heat distribution mapping. Pieces of ceramic were heated on a hot plate to a temperature of 36 °C and subsequently placed on different positions on the device to study the spatial resolution of the pixel array. The temperature differences observed by each of the pixels were deciphered from the captured voltage signals using a Seebeck coefficient S = 43.2 μV K^−1^, which corresponded to that of the film of equivalent thickness as presented in Table 1. The absolute temperature was then obtained by adding the temperature difference to the known temperature of the reference ground (~24 °C). In applications requiring absolute temperature values, the temperature of the reference ground must be either controlled or monitored separately. This can be realized by either positioning the reference ground to the periphery of the device or measuring the absolute temperature of the reference ground with an external sensor. The bottom row of Figure 3b presents the resulting temperature signals as two-dimensional natural neighbor interpolation heat maps, along with the corresponding photographs and infrared camera images on the top and middle rows, respectively. The sensor captured the location, shape, orientation and temperature of the object accurately.

Magnitude of the thermoelectric signals in the presented shape detection experiments was typically on the order of 250–370 μV, while thermal noise present in the signals was approximately 8–10 μV. Therefore, the resulting signal-to-noise ratio (SNR) in temperature-based shape detection was roughly 35. SNR was noted to vary depending on the relative temperature between the reference ground and the ceramic piece.

## 4. Discussion

Contrary to the thermoelectric energy harvesting devices conventionally evaluated via power factor or the thermoelectric figure of merit ZT = S^2^σT κ^−1^, where κ denotes thermal conductivity, sensing architectures exploiting the thermoelectric effect are poorly compared via these indicators. On the other hand, the thermoelectric sensor performance is largely determined by the Seebeck coefficient, which directs the future search for materials possessing both a high Seebeck coefficient and compatibility with liquid-phase processing.

To assess the performance of the sensor device in comparison with alternative carbon nanomaterials, large-area carbon nanomaterial films with thermoelectric parameters reported are compared to the multilayer graphene films studied in this work. Films made of semiconducting single-walled carbon nanotubes (SWCNTs) have been measured to possess S > 150 μV K^−1^ [47]. The use of SWCNTs is, however, restricted due to the more complex fabrication processes and the difficulty of separating the semiconducting species from the poorer performing metallic SWCNTs [47,48]. On the other hand, reduced graphene oxide films have been measured to have S ~ −90 μV K^−1^ [49], thus showing promise for sensing applications as described here. They are, however, again hindered by fabrication processes more complex than what is reported for the multilayer graphene films presented in this work.

When compared to results from previous studies on p-type liquid-phase exfoliated graphene films, the electronic properties agree well with atmospherically doped films [20]. However, in comparison to polymer-doped films, our results show a lower hole concentration but a larger carrier mobility [16]. A likely reason is that the absence of other than atmospheric dopants results in a lower charge carrier concentration, but at the same time, there are no large dopant molecules contributing to carrier scattering, which promotes higher carrier mobility. While the resistivities observed in this work range from 48 to 87 µΩ m, in sound agreement with similarly fabricated films [20], the values are an order of magnitude greater compared to those reported by Novak et al. [16], likely due to smaller flake size and, thus, shorter carrier mean free path and enhanced interface scattering. As a further comparison, the mobility values of inkjet-printed n-type graphene films have been reported to possess electron mobilities ranging from 8.9 to 23.9 cm^2^ V^−1^ s^−1^ as a function of increasing film thickness [50], agreeing well with the results reported here.

In applications of graphene requiring optical transparency, the desired performance is often achieved by reducing the film thickness, resulting in reduced electrical conductance [11]. Concurrently, the proposed sensing architecture is not significantly impaired by increased film resistance, which mainly manifests itself as thermal noise in the thermoelectric signal [6]. Thus, the device benefits from the open-circuit configuration and passive sensing principle.

Ink formulation similar to the one utilized here has previously been optimized for inkjet printing [20], paving the way for the integration of printed thermoelectric elements as parts of more complex flexible electronic configurations. Inkjet printing may also provide remarkable advantages in fabricating devices such as those presented here, particularly in the realization of readout leads and the corresponding electrical insulation [51].

The demonstrated large-area thermoelectric performance of ink-based graphene films suggests that the concept could be scaled to suit fabrication methods such as gravure and flexographic printing, both of which have already been experimented with 2D material-based inks [52,53]. In addition to pursuing different deposition methods for the thermoelectric layer, alternative strategies for the realization of the dielectric layer could further improve the scalability of device fabrication. Potential candidates could include, for example, ink-based approaches comprising hexagonal boron nitride (hBN) [54] or flexible ion-gel dielectrics suitable for low-temperature fabrication, which have previously been exploited in conjunction with graphene [55].

## 5. Conclusions

In summary, a large-area thermal distribution sensor utilizing a thermoelectric film deposited from low-cost multilayer graphene ink was presented. The inexpensive and facile sensor configuration could detect thermal input from intermittent heat sources, such as human touch, as well as detect objects based on temperature distribution mapping. The finite-element model utilized in studying the device performance provides an accurate reproduction of the concept and allows for optimization of device parameters for the application at hand. The applied fabrication processes are readily scalable into high volumes, which, together with the simple device architecture, offers an accessible route to using multilayer graphene for thermoelectric sensing in application areas such as wearable and flexible electronics.

## Figures and Tables

**Figure 1 sensors-20-05188-f001:**
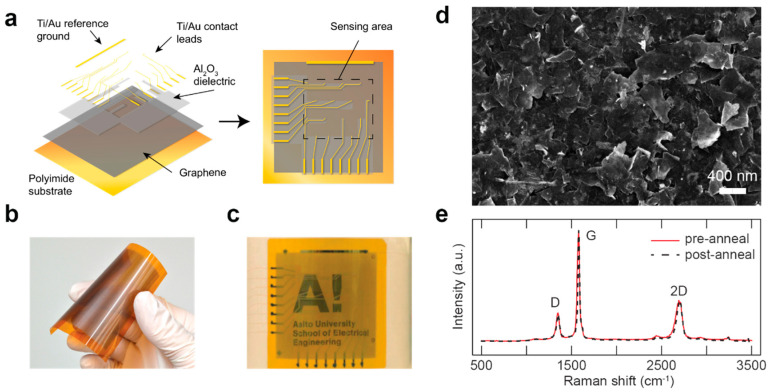
(**a**) Exploded-view drawing and top view of the device architecture. Graphene ink is deposited on a flexible polyimide substrate, followed by the deposition of Al_2_O_3_ dielectric layer and Ti/Au readout leads. Graphene film together with the readout leads forms a four-by-four pixel thermal sensing array. (**b**,**c**) Graphene film on polyimide substrate before (**b**) and after (**c**) the deposition of Al_2_O_3_ and Ti/Au layers. In (**c**), the device was attached onto the outer surface of a beaker, demonstrating both the flexibility and transparency of the sensor. (**d**) Scanning electron microscope (SEM) image of the device surface showing the stacked multilayer graphene flakes. (**e**) Raman spectra of the graphene flakes before (**black line**) and after (dashed **red line**) annealing. The spectrum remained largely unaffected by the annealing treatment.

**Figure 2 sensors-20-05188-f002:**
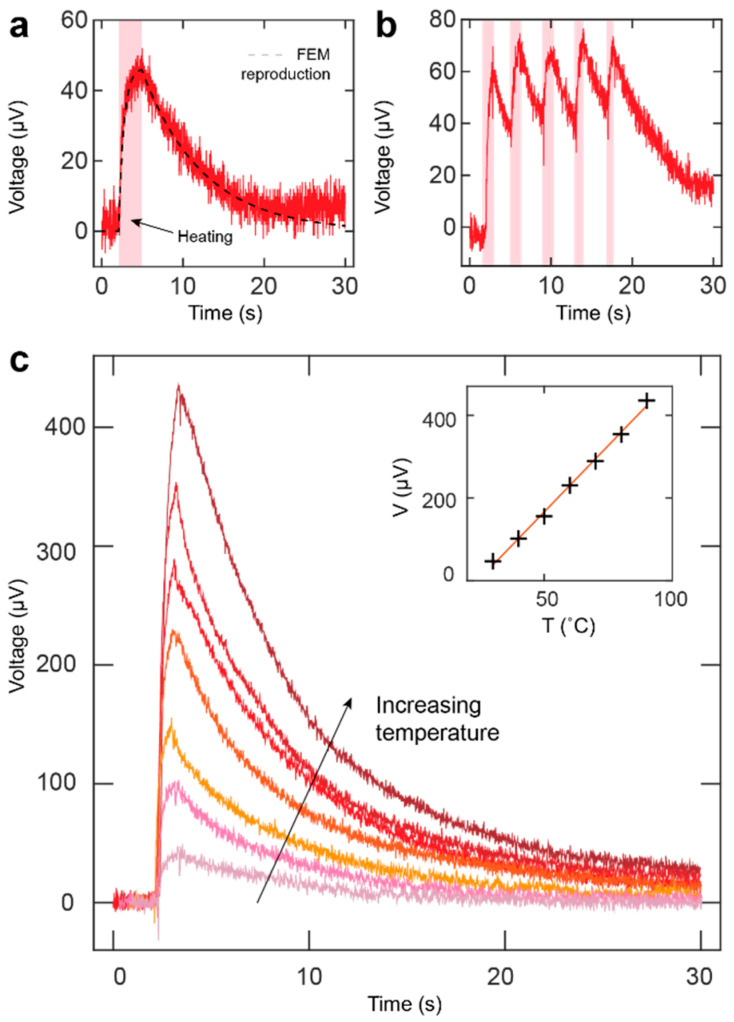
Transient responses of the fabricated device. (**a**) Single touch on a single channel. The duration of heating is highlighted as a shaded region. (**b**) Consecutive touches on a single channel. (**c**) Single channel responses to seven different temperatures ranging from 30 °C to 90 °C in 10 °C intervals. The inset shows the linear trend in signal peak voltages at different temperatures.

**Figure 3 sensors-20-05188-f003:**
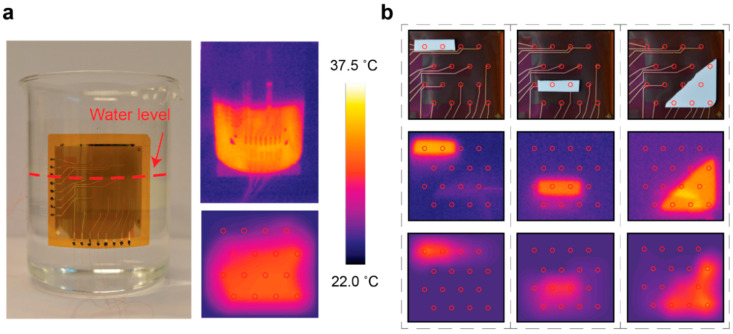
(**a**) Device conformally attached onto a beaker surface with a radius of 5.25 cm along with a temperature distribution map of the beaker partly filled with 36 °C water. (**b**) Device on a flat surface with heated ceramic pieces on the active area (**top row**) along with heat distribution maps produced by infrared camera (**middle row**) and the presented thermal distribution sensor (**bottom row**). The pixel locations in the four-by-four matrix are highlighted as red dots.

**Table 1 sensors-20-05188-t001:** Electronic and thermoelectric properties for ultrasonic-assisted liquid phase exfoliation (UALPE) multilayer graphene films of different thicknesses.

Sample	Thickness [nm]	Seebeck Coefficient [μV K^−1^]	Sheet Resistance [kΩ sq^−1^]	Power Factor [μW m^−1^ K^−2^]	Hole Mobility [cm^2^ V^−1^ s^−1^]	Hole Concentration [10^19^ cm^−3^]
4 mL	78.2	44.7	1.12	22.9	33.3	3.0
8 mL	108.5	43.2	0.78	22.1	35.3	3.1
12 mL	214.8	40.1	0.26	28.5	38.1	2.9
16 mL	277.1	37.4	0.17	29.3	40.8	2.6

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
