# Peer review of "Large-Area Thermal Distribution Sensor Based on Multilayer Graphene Ink"

_sensors, 2020, doi:10.3390/s20185188_

Round 1

Reviewer 1 Report

In this manuscript, the author demonstrates a large‐area thermal sensor based on distributed thermocouple architecture and ink‐based multilayer graphene film. This thermal sensor is equipped with a common reference ground and 16 readout leads, which is able to detect the temperature as well as its distribution through Seebeck effect. From my point of view, the work provides interesting results to the application in wearable sensors. The manuscript is recommended for publication after addressing the following comments: 1. As the thermal sensor is working based on Seebeck effect, In order to calculate the accurate temperature at the sensing point, the temperature at reference ground should be either same with the environment temperature or at a fixed known temperature which is lower than that at the sensing point. However, for practical application such as wearable sensor or the water temperature sensing experiment performed in the manuscript, the temperature at the reference ground can be easily affected by the heat source (such as skin or the water vapor), resulted in inaccurate read out value. It is better to further explain how to avoid such issue. 2. The thermal sensor is composed of multilayer graphene, as the graphene flakes structure can be changed during bending, as a result, the Seebeck coefficient may also change. The author should provide a test value of Seebeck coefficient for the sensor under different curvature state

Reviewer 2 Report

The manuscript is well written and interesting, and it can be of wide interest in the field of layered graphene-based sensors. However, I believe that some analytical and experimental implementations are needed. Specifically, here there are some aspects that need to be improved:

1). Raman spectra before and after heating to 373°C are shown in Figure 1e. These spectra clearly show that the graphene of the film remains substantially unchanged after heat treatment from characteristic bands (D-band, G-band, 2D-band). It would be opportune to determine the thickness of the graphene layers from the I2D/IG ratio and from the shape of 2D band.

2). It would be also useful to see how these characteristics (i.e. Raman spectra) change in the series of films obtained from 4, 8, 12 and 16 ml (and not only that of 8 ml solution).

3). The electrical resistivity of the films corresponds to about 0.0048-0.0087 Ohm cm (Table 1). However, for films it is more usual to evaluate in Ohm/sq. I suggest converting the resistivity values considering the geometric aspects of the films.

4). Also, in Table 1, together with the resistivity values, hole mobility, and concentration values at room temperature are shown. In the discussion, a comparison with values provided by the most representative papers in the literature about graphene films would be desirable.

5). I believe that there is also a relationship with the deformation of the sensor that needs to be determined. In addition to the touch characteristics shown in Figure 2, it would, therefore, be appropriate to have information about the sensor when it is bent.

6). Finally, since the use as a water level sensor is shown, I think it is important to evaluate the electrical characteristics under controlled humidity conditions.

Reviewer 3 Report

Authors describe the fabrication and characterization of a temperature sensing device. In some parts information is lacking and the authors should integrate the description and discussion of the results as indicated in the comments.

Row 104, figure 1a represent the device architecture. It is unclear if the elevation of the readout legs represents the temperature. In this case, I suggest to separate structure and thermogram in two separated figures and add information to the caption.

Row 110: authors state: “When the contact between a readout lead and the graphene film is heated, a temperature gradient forms between the contact point and the reference ground, resulting in a thermoelectric voltage determined by the effective Seebeck coefficient of the graphene film and the readout lead”. The authors fabricated a matrix of temperature sensing legs on a 6x6 cm of graphene films. When an object ( a finger) touches the surface, the increase of temperature is caused not just in a point but on a rather large area and the resulting voltage likely depends on the whole contacts heated not just their end. Why the authors do not deposit an Al2O3 coating with a matrix of holes where the TiAu get in touch with the underlying graphene to get T measurements just at the end of the readout legs?

Row 121: Raman. The authors just observe that the Raman spectra reveal that the crystalline nature of the graphene coating is preserved. This is likely due to the multilayered structure of the initial graphene flakes. The SEM image shows a rather disordered structure on the macro-scale. The process of reduction seems to not induce any variation in the Raman spectrum although a higher number of defects should be observed. The multilayered structure of the graphene flakes is also visible analysing the 2D band. A more detailed analysis of the Raman spectra is required.

Row 140: thermoelectric properties. A comparison of the thermoelectric properties of the 16 legs should be given/commented. Do the different legs behave in a similar way? Can the authors estimate the error done measuring the same temperature with different legs?

Row 155: authors state: “However, thinner films appear preferential for sensor application due to both higher transparency and Seebeck coefficient”. Why transparency is important for sensing temperature?

Row 222: Discussion. Authors should compare the performances of their sensing device with other based on C nanostructures such as CNT or similar. As in the case of GO, also C nanostructure can be deposited on surfaces in thin layers. Then authors should justify GO as the material of choice for fabricating the sensing device.

Round 2

Reviewer 1 Report

The author has addressed all my comments and questions, the manuscript is suggested to be published.

Reviewer 2 Report

The manuscript has been properly revised and can be accepted for publication.

Reviewer 3 Report

//